# Hybrid CNN-SVR Blood Pressure Estimation Model Using ECG and PPG Signals

**DOI:** 10.3390/s23031259

**Published:** 2023-01-22

**Authors:** Solmaz Rastegar, Hamid Gholam Hosseini, Andrew Lowe

**Affiliations:** 1PHD Media, Auckland 1024, New Zealand; 2Institute of Biomedical Technologies, School of Engineering, Computing and Mathematical Sciences, Auckland University of Technology, Auckland 1010, New Zealand

**Keywords:** electrocardiogram (ECG), photoplethysmography (PPG), convolutional neural network (CNN), continuous blood pressure

## Abstract

Continuous blood pressure (BP) measurement is vital in monitoring patients’ health with a high risk of cardiovascular disease. The complex and dynamic nature of the cardiovascular system can influence BP through many factors, such as cardiac output, blood vessel wall elasticity, circulated blood volume, peripheral resistance, respiration, and emotional behavior. Yet, traditional BP measurement methods in continuously estimating the BP are cumbersome and inefficient. This paper presents a novel hybrid model by integrating a convolutional neural network (CNN) as a trainable feature extractor and support vector regression (SVR) as a regression model. This model can automatically extract features from the electrocardiogram (ECG) and photoplethysmography (PPG) signals and continuously estimates the systolic blood pressure (SBP) and diastolic blood pressure (DBP). The CNN takes the correct topology of input data and establishes the relationship between ECG and PPG features and BP. A total of 120 patients with available ECG, PPG, SBP, and DBP data are selected from the MIMIC III database to evaluate the performance of the proposed model. This novel model achieves an overall Mean Absolute Error (*MAE*) of 1.23 ± 2.45 mmHg (*MAE* ± *STD*) for SBP and 3.08 ± 5.67 for DBP, all of which comply with the accuracy requirements of the AAMI SP10 standard.

## 1. Introduction

Blood pressure (BP) is one of the most important vital signs of the human body that can be assessed as a risk factor for severe health conditions, especially cardiovascular diseases (CVD) and hypertension. An accurate, continuous, and cuff-less BP monitoring technique could help clinicians to improve the rate of prevention, detection, and diagnosis of hypertension and manage related treatment plans. 

Many factors, such as various abnormalities in cardiac output, blood vessel wall elasticity, circulated blood volume, peripheral resistance, respiration, and emotional behavior, influence BP. Notably, the complex and dynamic nature of the cardiovascular system necessitates that any BP monitoring system should benefit from intelligent technology that can extract and analyze compelling BP features. Therefore, employing traditional BP measurement methods and using handcrafted features for continuous BP monitoring is cumbersome and computationally intensive.

BP measurement through engineered feature extraction has been used by many researchers recently. It is considered that BP is the result of various physiological and neurological factors, so including feature extraction in a BP estimation model could significantly improve the accuracy of the measurement [1,2,3]. The most common feature used to develop a BP measurement model is Pulse Transit Time (*PTT*). The relationship between BP and *PTT* has been well studied [4,5], and *PTT* is recognized as a valid and well-accepted method for estimating BP [6,7]. *PTT* has been used to develop wearable devices for continuous BP measurement, as it can be conveniently estimated from ECG and PPG signals [8]. Moreover, due to the simplicity of this method, it can be used for a wide range of patients regardless of age or underlying conditions. 

However, the methods that rely on hand-engineered feature extraction have two considerable limitations. The first issue is that calculating several features at once is difficult due to individual-specific waveforms and the effect of motion artifacts. The other one is extracting desired features creates cost challenges in real-time monitoring, and it is also time-consuming. Brain-inspired mathematical models, known as Artificial Intelligence (AI), have been used as primary tools in machine learning. They are one of the most promising techniques for recognizing patterns of highly complex functions and emulating the non-linear relationship between inputs and outputs of non-linear systems. The deep learning class of machine learning algorithms has shown promising results in biomedical technologies such as risk assessment for hypertension [9] and echocardiography image analysis [10].

Deep learning techniques could be applied to BP feature extraction and classifying other physiological signals. Moreover, these techniques could be employed for estimating BP as a potential approach for achieving continuous BP measurement. This study aims to develop an optimized convolutional neural network (CNN)-based model to estimate BP using the fundamental capability of CNN algorithms to perform automatic feature extraction. A novel hybrid system that combines the optimized CNN model with the SVR model is proposed, where CNN is used as a trainable feature extractor, and SVR is used as the regression operator. The proposed hybrid CNN-SVR model can effectively learn more in-depth features of electrocardiogram (ECG) and photoplethysmography (PPG) signals and their corresponding systolic blood pressure (SBP) and diastolic blood pressure (DBP). Based on the flexibility of CNN architecture to deal with various inputs (ECG and PPG) that can be collated and obtained easily, the proposed model is suitable for wearable devices.

## 2. Materials and Methods

### 2.1. Database

The data in this study were obtained from the Medical Information Mart for Intensive Care (MIMIC-III) Waveform Database Matched Subset [11]. This database included multiple physiologic waveforms and numeric time series of vital sign measurements. It contained 22,317 waveform records and 22,247 numeric records collected from 10,282 distinct Intensive Care Unit (ICU) patients that were matched and time-aligned with MIMIC-III Clinical Database records.

The database provided high-resolution waveform data recordings at a sampling frequency of 125 Hz and clinical information on patients that had been hospitalized in the ICU from 2001 at the Beth Israel Deaconess Medical Centre. Each waveform record contained numerical data extracted minute by minute physiologic parameters. Data were obtained from patients aged 16 years or older, with a mean of 65.8 years, 44.1% of subjects were female, and 55.9% were male. Each subject could have several recordings with different time lengths, from seconds to a few hours.

An initial list of patients with available ECG, PPG, SBP, and DBP using the Physionet Bank tool was created. Patients with empty or a minimal number of samples were then removed from the list. To download the selected patients’ data in MATLAB, we developed an algorithm using the WFDB software package, the *wfdb2mat* function [12]. The following steps were taken to synchronize and align the data for each patient.
The duration of recording was specified and based on the requirement of this study; seven minutes of data were collected for each patient.Most of the missing samples occurred within the first minute of recording; all records were collected after one minute of the actual recording time.A 30 s gap was allowed between each one-minute collection interval to avoid recording consecutive pulses.Five ECG signals from leads I, II, III, AVR, and V were available during portions of some ECG records in the MIMIC III waveform database. However, these five ECG signals were not all available simultaneously. We employed the ECG lead AVR in this study due to better signal quality and availability in most records. The ECG and PPG waveforms were loaded into the MATLAB environment as a matrix with two labelled columns.The corresponding SBP and DBP specified as ABP Sys and ABP Dias, respectively, were loaded as a matrix with three labeled columns.Both matrices of steps 4 and 5 were concatenated together and saved in CSV format for training and testing the network.As each cardiac cycle interval was considered to be 0.6 s to 1 s, the signals were segmented into the cardiac cycle with a length of 1 s as one beat [13].

A summary of the above steps is illustrated in Figure 1.

Considering all the above criteria, the data from 120 patients were collected, with a duration of seven minutes of data for each patient. A total of 420 samples of ECG, PPG, SBP, and DBP were individually for each subject and, collectively, 50,400 samples (equal to 14 h of training data for each signal) were extracted.

### 2.2. Pre-Processing

The ECG signal is one of the crucial inputs to the BP estimation algorithm, and a clean and pre-processed ECG signal will increase the accuracy of BP measurement performance. The following procedure was utilized to pre-process and denoise the ECG signal.

Generally, baseline wandering is the noise artifacts that affect ECG signals. The source of these artifacts is often respiration and usually lies between 0.15 and 0.2 Hz. Removing baseline wander is necessary to reduce the irregularities in beat morphology [14]. In this study, the baseline wander of the ECG waveform was eliminated by shifting the signal baseline, aiming to represent the correct amplitude. To remove the trend, a low-order polynomial was fitted to the signal, and the polynomial was used to detrend it. ECG signals are often noisy due to random body movements, poor contact with electrodes, and electrical activities of other muscles in the body [15]. However, standard filters cannot remove the noise sufficiently, as an ECG is a non-stationary signal [16].

There are several methods used for denoising signals, including the wavelet transform [17,18,19], Savitzky–Golay filtering [20,21], and adaptive filtering [22]. In this study, the Savitzky–Golay filtering was used to remove the ECG signal noise as it has previously shown better performance compared with other commonly used denoising methods [15]. This method is based on a local least-squares polynomial approximation, which was first proposed by Savitzky and Golay [23]. The sgolayfilt function in MATLAB was utilized to implement the Savitzky–Golay smoothing filter. This function computes the smoothing polynomial coefficients, delays the alignment, and performs the transit effects at the start and end of the data record. An example of a noisy and denoised ECG signal is shown in Figure 2.

### 2.3. R-Peak Detection

As mentioned in the introduction, *PTT* is one of the features that has shown a high correlation with BP [24]. *PTT* is related to PWV, and both exhibit an inverse relationship with BP [8]. The *PTT* is the time taken for an ECG R-wave to propagate to the PPG sensor located at the patient’s finger, as shown in Figure 3 [25].

Thus, all critical points related to *PTT* need to be available in each cardiac cycle sample. For this purpose, the R-peak of ECG signals was detected after the pre-processing step. Using the detected index, the length of each cardiac cycle centered on R-peaks was calculated. The same approach was taken to align all the samples, and the identical length was used to resample the PPG and BP signals.

In this study, the thresholding method was selected for peak detection. The ECG and PPG signals were normalized based on their corresponding value range to optimize signal amplitude changes and reduce peak detection error. The R-peak of ECG signals was detected by thresholding peaks above 0.5 mV. Moreover, to avoid detecting unwanted and false peaks, the minimum peak distance was set by 125 samples. According to the MIMIC III database, the sampling frequency was 125 Hz, corresponding to one cardiac cycle of 60 BPM. We considered the ECG signals with a heart rate of greater or less than 60 BPM by locating the R-wave in the center of the window and eliminating samples from the tail of the signal for cases of less than 60 BPM or zero padding for cases greater than 60 BPM.

The indexes related to the R-peak were used to resample the cardiac cycle (centered on the R-peak) and aimed to include all the *PTT*-related points. Figure 4 illustrates a sample of the peak detection algorithm for an ECG signal.

### 2.4. Hybrid CNN-SVR Model

The proposed system was designed to integrate CNN and the SVR networks. The theory of CNN is briefly introduced in Section 2.4.1, the CNN Model for BP Estimation is presented in Section 2.4.2, and the SVR structure is discussed in Section 2.4.3. The architecture of the proposed hybrid CNN–SVR model is presented in Section 2.5, followed by experimental results and comparison.

#### 2.4.1. Convolutional Neural Networks

Considering the fact that the convolutional neural network (CNN) is the most popular technique used in deep learning [26], we found that a CNN-based architecture could be investigated and applied to continuous and cuff-less BP monitoring using the ECG and PPG signals. One of the significant benefits of this approach is the ability of CNNs to perform perception tasks, which allows them to learn the BP-relevant features from ECG and PPG signals. Therefore, the first step would be choosing a suitable network architecture and training it with ECG and PPG signals. The last layer of the network would be a regression layer to estimate BP as the output of the proposed model.

The building block of the CNN contains multiple layers, such as convolution layers, pooling layers, and fully connected layers. CNN can have hundreds of layers, and each of them could extract different features. The network architecture can vary depending on the nature of the input data and the related application. The input data generally include several localized features arranged as several feature maps. The multiple of a convolution matrix and a filter matrix is called the feature map. After applying each convolution and pooling layer, the size of the feature maps becomes smaller. After combining the features across all frequency bands, one or more fully connected hidden layers are added before feeding them to the output layer after the final CNN layer.

The main building of CNNs is the convolution layer, which computes the number of learnable filters and the dimension of input data (width, height, and depth). After the convolution layer, the pooling layer is typically used for down-sampling and reducing the number of computations and overfitting. The pooling operation can be max pooling and average pooling. Max pooling computes a maximum value from the region overlapped by the kernel, whereas average pooling gives an average of all pixel values from the overlapped region.

The extracted feature maps are then sent to activation functions that conduct non-linear transformations. Non-linear activation functions are important because, otherwise, the network would be a linear predictor without the ability to learn non-linear features. The ReLU layer replaces all negative values with zeros received as inputs.

During the training of the deep neural network, the data distribution of each layer must be constantly changed to prevent early saturation of the non-linear activation function in the whole network. Accordingly, we utilized batch normalization in the CNN structure to reduce the internal covariate shift, thereby avoiding the vanishing gradient problem and accelerating the training of the CNN network. This had a beneficial impact on the regularization of the CNN model and the reduction in dropout utilization. Finally, as the name suggests, in the fully connected layer, all neurons in this layer are connected to all the neurons in the previous layers. 

Following the success of CNN in the fields of speech recognition, emotion recognition, and other speech analysis applications [27,28,29], a novel CNN-based BP estimation is proposed in this study. The proposed method can detect the knowledge related to BP estimation from ECG and PPG signals by extracting complex BP-related features automatically instead of designing them. The engineered features are generally not robust, due to noise, scaling, and displacement; however, the extracted features’ quality significantly affects the network’s performance. The proposed method can overcome the drawbacks of feature recognition of the *PTT*-based techniques and the low BP prediction accuracy of the non-*PTT*-based methods.

The proposed deep learning models in this study were trained on the ECG and PPG signals collected from the MIMIC database. By utilizing the advantage of deep learning, which automatically extracts features, ECG and PPG signals were used intact to reflect the inherent characteristics of the signals themselves. Notably, the BP estimation was relatively accurate without individual calibration.

The design of CNN architecture is challenging and involves several aspects, including the performance metric, loss function, optimization, and hyperparameter setting. The hyperparameter setting includes the number of hidden layers and channels for each layer, the pooling layer, the batch normalization, the activation layers, the learning rate, and batch size [30]. Although there are various pre-trained CNN networks, such as AlexNet, GoogLeNet, and ResNet, this study utilized a novel CNN network designed to suit the complexity of the selected training dataset. Creating a network required determining a network configuration to provide the most control over the network and optimizing the results.

The proposed CNN structure was built using five convolutional layers. Each convolutional layer was followed by batch normalization, an activation layer, and an average-pooling layer. The input data were fed into the convolutional layers with filter size W × H and kernel size N, where W and H are the filter width and height, respectively, and N is the filter depth.

The batch normalization was implemented for regularization after each convolutional layer [31]. After batch normalization, the average-pooling layer was used to extract high-level features [32]. The extracted features were then sent to the ReLU layer that performs the non-linear transformation. The dropout layer was then applied to avoid overfitting, which causes the deep learning network to fail in predicting additional or future data. After all the convolutional layers were implemented, the outputs were fed to the fully connected layers. The fully connected layer multiplied the input by a weight matrix and added a bias vector.

The output layer is a regression layer that computes the half-mean-squared-error loss for regression problems. The mean square error (*MSE*) is given by Equation (1):(1)$$MSE=\overset{R}{\underset{n=1}{\sum }}\frac{{\left(t_{i\;}-\;y_{i\;}\right)}^{2}}{R}$$
where $t_{i}$ is the estimated BP, $y_{i}$ is the actual value, and R is the total number of values in the testing dataset.

#### 2.4.2. CNN Model for BP Estimation

The proposed CNN architecture had five convolutional layers, which were the core of the network, followed by a fully connected layer with two neurons and a regression layer to address the regression problem. The last two layers were developed for calculating the SBP and DBP. Overfitting is a common problem for a network with high variance. To reduce the training data’s overfitting and improve the network’s performance, a regularization layer with a dropout ratio of 20% was considered before the fully connected (FC) layer. The details of the proposed architecture for an optimized CNN are presented in Figure 5b.

#### 2.4.3. SVR Model

Different models can be used to solve classification, regression, and abnormality detection problems. A support vector machine (SVM) is one method that can overcome these problems. An SVM encompasses a range of different models divided into linear non-separable vector machines, linear separable SVMs, and non-linear SVMs. Using the Kernel function, the non-linear SVM converts linearly non-separable problems into linearly separable problems from low-dimensional space to high-dimensional space and then linearly solves non-linear issues [33]. The SVM algorithm can be employed for classification or solving a progression problem. The regression or the function approximation version of SVM is referred to as support vector machine regression (SVR) [33].

The power of this model to deal with the non-linear relationship of input and output data makes it suitable to address the non-linear relation between the features extracted from ECG and PPG signals and the actual BP. Therefore, an SVR model using complex characteristics of the human physiological index is proposed to estimate SBP and DBP.

##### Kernel Function Selection

In some cases, the data are not linearly separable, so the kernel function is used to map the projection of input space to a higher dimension where a linear separation is feasible. The goal is that after the transformation to the higher dimensional space, the classes become linearly separable. Therefore, support vector machine models can use the kernel function K(x, x′) to establish a non-linear support machine. Different kernel functions produce different algorithms for mapping the data into different dimensional spaces. The most common kernel functions are the polynomial kernel functions, Gaussian Radial Basis Function (RBF), and sigmoid kernel function [33].

The RBF kernel was adopted in the SVR algorithm by mapping the original feature space x=(PTT, HR, …) onto the new feature space $x\prime =\left(x_{1},x_{2},x_{3},\;\ldots x_{n}\right).$ Therefore, the new set of the BP indicator data was expressed by a linear regression formula in the feature space to determine a non-linear mapping model between ECG and PPG signals and BP. The SVR model did not establish based on a polynomial kernel function and a sigmoid kernel function. The polynomial SVR model needs a prolonged parameter optimization process to converge, and the sigmoid SVM model has fast parameter optimization, but the predictions are poor. Moreover, fewer parameters are required for the RBF kernel than the polynomial kernel; the number of parameters can contribute to the complexity of the overall system.

##### Description of the Proposed SVR

SVR has a unique ability to solve the non-linear regression problem. The non-linear mapping φ(x) mapped the input sample *x* into a high-dimensional feature space and then estimated the regression function through a linear model built in this feature space. The main idea was to use non-linear mapping to map the input space onto a high-dimensional feature space. The non-linear model is shown in Equation (2) [34]:(2)f(x,ω)=ω·φ(x)+b
where x=(PTT, HR, …),  ω is the weight vector, and b is the threshold. The *ω* and *b* can be obtained through the following optimization equation (Equation (3)).
(3)$$\min\frac{1}{2}\;∥\omega {∥}^{2}\;+c\;\overset{1}{\underset{i=1}{\sum }}\;(\xi_{i\;\;}+\;\xi_{i}^{*}),$$
is subject to:$$y_{i}-\left(\omega^{T}x_{i}+b\right)<\varepsilon +\xi_{i}$$
$$\left(\omega^{T}x_{i}+b\right)-y_{i}<\varepsilon +\xi_{i}^{*}$$
$$\;\xi_{i}+\xi_{i}^{*}\geq 0$$
where C is a penalty factor, ε is the loss function, and $\;\xi_{i\;}$ and $\xi_{i}^{*}$ are different relaxation factors. Equation (4) is as follows:(4)$$f\left(x\right)=\;\overset{l}{\underset{i}{\sum }}(-\alpha_{i}+\;\alpha_{i}^{*})\;K\;\left(x_{i},\;x\right)+b,$$
where $\alpha_{i}\;$ and $\alpha_{i}^{*}$ are Lagrange multipliers, l is the number of SVs, and $K\left(x_{i},\;x\right)$ is a kernel function.

After comparing different kernel functions, the RBF kernel was chosen to be used in this study. Unlike the linear kernel, the RBF kernel transforms the database into a non-linear high-dimensional space, making it possible to overcome the non-linear relationship between features and BP. In addition, compared with the polynomial kernel, it has less model complexity due to fewer tuning parameters. The RBF is shown as Equation (5):(5)$$K\left(x_{i},\;x\right)=\exp(\;-\gamma \;∥x-x_{i}{∥}^{2})$$
where γ is the kernel parameter. The kernel parameter, gamma (γ), can adjust the influence of a training sample. The larger value will decrease under the influence of the training sample.

Considering Equations (5) and (6), selecting the proper value for C (penalty factor), gamma (γ), and ε (loss function) can effectively increase the accuracy of the SVR model. The C parameter can balance the prediction error in the training set [35]. Additionally, a smaller C makes the decision surface smoother, whereas a larger  C allows the model more freedom to use more samples as support vectors so that all the training samples can be accurately classified or predicted. A larger Gamma, γ, has less influence on the training sample.

### 2.5. The Architecture of the Proposed Hybrid CNN-SVR Model

The proposed hybrid CNN–SVR model was designed by replacing the last output layer of the CNN model with an SVM classifier. This model employs the optimized CNN model to extract robust features from the time series of ECG and PPG signals and present them to the SVR network. The last output layer of the proposed CNN is replaced with an SVR model to design the hybrid CNN-SVR model that consists of the following three stages:

First, the time series of ECG and PPG signals were directly fed into the proposed CNN model for training the CNN network. We used lead AVR for ECG and PLETH for the PPG signal in the MIMIC III waveform database as parallel input to the network and concatenated them vertically.

Secondly, the CNN network was employed to extract representative features from the time series of ECG and PPG signals. 

Finally, the extracted features by the CNN network were applied as input to the SVR for BP estimation. The proposed hybrid CNN-SVR method did not use any engineered feature extraction techniques for the ECG and PPG signals.

The structure of the proposed hybrid CNN-SVR model is shown in Figure 6.

For feature extraction with CNN, five convolutional layers are used. Each convolutional layer is followed by batch normalization, ReLU, and average pooling layers. Batch normalization can robustly handle overfitting and underfitting problems. A dropout regularization technique with a dropout ratio of 20% is employed to reduce overfitting before the fully connected layer.

The last layer in the CNN model is replaced with an SVR with an RBF kernel. Then, the SVR takes the outputs from the hidden layer as a new feature vector for training. Once the SVR is well trained, it performs the regression step and predicts SBP and DBP with automatically extracted features from CNN. In this case, SVR can handle high-dimension regression problems better than the simple linear combination. Furthermore, the combination of CNN and SVR networks can benefit from the advantages of each layer and enhance the accuracy of the estimation

## 3. Model Training and Experimental Results

The implementation of the proposed hybrid CNN-SVR is shown in Figure 7.

To evaluate the feasibility of the hybrid CNN-SVR model for BP measurement, an experiment was conducted on 120 subjects from the MIMIC-III database. The data related to 70% of the selected subjects were assigned to the training set, and the rest were assigned to the testing set. The CNN network of the CNN-SVM combined model was trained with the Stochastic gradient descent [36] optimizer with an initial learning rate of 0.001 and a piecewise drop of 0.1 for every epoch. The training process was conducted with 30 epochs, as a higher value results in longer training time without any changes in performance. The mini-batch size was set to 60 as it has been observed in an experiment that a larger batch size leads to less time to train and poorer generalization, and less accuracy. 

To establish an SVR model to predict the SBP and DBP using the optimal parameters C and γ, the parameters were set to (100, 10) for C, (100, 10) for γ, and [0.01,1] for ε. These values were chosen based on the promising results of Zhang et al. [37], which were achieved by the optimization function created on 10-fold cross-validation. The proposed SVR model was implemented in the MATLAB environment.

### 3.1. Accuracy Performance

To evaluate the BP estimation accuracy of the proposed model, four measurement metrics, including accuracy, *RMSE*, the total mean absolute error (*MAE*), and total standard deviation error (*STD*), as shown in Equation (6) to Equation (9), were selected.
(6)$$Accuracy\;=100\;\times \;\frac{x}{y}$$
where x is the number of correct estimations and y represents the total number of model estimations values.
(7)$$RMSE=\;\sqrt{\frac{\sum_{i=1}^{n}\left(x_{i}-\;y_{i}\right)}{n}}$$
(8)$$MAE=\;\frac{\sum_{i=1}^{n}\left|x_{i}-\;y_{i}\right|}{n}$$
(9)$$STD=\;\sqrt{\frac{\sum_{i=1}^{n}{\left(x_{i}-\;y_{i}\;-ME\right)}^{2}}{n-1}}$$
subject to:(10)$$ME=\frac{\sum_{i=1}^{n}\left(x_{i}-y_{i}\right)}{n}$$
where *ME* is the mean error, $x_{i}$ is the estimated BP, $y_{i}$ is the actual value, and n is the total number of values in the testing data.

Figure 8 shows the average results of the actual and the predicted SBP and DBP for all patients. It shows that the hybrid CNN-SVR model for SBP and DBP estimation generates values close to the actual values, with the highest estimation accuracy among all the proposed methods.

The accuracy of the proposed method was calculated by dividing the number of correct estimations by the actual number of data samples. The average accuracy of 97.83% for SBP and 93.47% for DBP across all subjects was achieved. Moreover, *RMSEs* of 1.89 mmHg and 3.91 mmHg for SBP and DBP, respectively, were calculated using hybrid CNN-SVR.

Table 1 shows the results of the proposed hybrid CNN-SVR model against the AAMI standard. The AAMI standard requires a study population of at least 85 subjects, while the proposed study included 120 subjects for evaluating the results. In addition, the AAMI requires a mean absolute error (*MAE*) of less than 5 mmHg for non-invasive BP measurement and a standard deviation error of more than 8 mmHg.

It can be seen from Table 1 that the hybrid CNN-SVR model fully satisfies the AAMI criteria for non-invasive and continuous SBP and DBP measurement.

### 3.2. Comparison of the Results with the Related Works

A comparison between the performance of the proposed CNN-SVR BP model and the related work is shown in Table 2. To achieve a fair comparison, the studies that reported their results with the same evaluation metric and database were considered in this study. The proposed study using the combination of CNN and SVR to predict BP achieved better performance in comparison with the selected studies. The *MAE* for SBP was significantly lower, and the *MAE* for both SBP and DBP were within the AAMI standard criteria.

Kachuee et al. [38], Slapničar et al. [39], Chen et al. [40], Khalid et al. [41], Ertuğrul et al. [42], and Wang et al. [43] proposed a feature-extraction-based technique to estimate BP. Although some of these studies reported acceptable results [40,41,42,43], they still used various time-consuming and complex feature extraction techniques. Moreover, two studies reported on employing CNN-based techniques without feature extraction [44,45]. However, the error results and the number of subjects were not within the required BP standard. The CNN-based model [45] was chosen for this comparison and used a sufficient number of subjects for the experiment. Nevertheless, this method used the calibration technique, and the error margin was not acceptable by BP standards.

Looking at individual related works in Table 2, Kachuee et al. [38] proposed a method that employs pre-processed physiological features and a machine learning model using the *PTT* approach and calibration. The database was divided into training and test sets based on the number of subjects that showed promising results, according to the British Hypertension Society (BHS). Slapničar et al. [39] used the deep-learning spectro-temporal ResNet and a large dataset to estimate the BP. The PPG features used to train and test the proposed model and divided based on the number of subjects achieved reasonable accuracy; however, just the PPG signal was used to estimate the BP.

Chen et al. [40] used the SVR model and fourteen features from ECG and PPG signals to estimate the BP. The accuracy of BP estimation increased significantly. The database was randomly divided into the training and test set, which might cause the overlapping of the subjects in the training and testing datasets. Khalid et al. [41] employed three PPG features to train and test different machine-learning algorithms to find the best model to estimate BP. A small database was used to evaluate the models; the training data included the old subjects, while the test dataset included young subjects. Although the proposed model achieved promising results, the study was limited to 32 subjects and hand-engineered PPG features. Ertuğrul et al. [42] presented an extreme learning machine (ELM) model using the time–frequency domain and statistical features. The results were within the standard criteria, but they used minimal data (only five subjects), and the training and testing database was divided randomly. In [43], Wang et al. conducted the BP estimation using ANN and the multitaper method (MTM) to extract the BP-related features. However, the database was randomly split into training and testing datasets.

The hybrid CNN-SVR BP model proposed in the study was estimated with higher accuracy and the least error while avoiding overlapping the subjects of the training set with those of the test set by dividing them based on the number of subjects. The proposed hybrid CNN-SVR BP model is the first CNN and SVR combination that targets BP estimation to the best of our knowledge. This novel hybrid model fills the research gaps by utilizing CNN, which can extract robust features automatically instated of using the traditional feature extraction approaches. Then, SVR reconstructs the non-linear relationship between the in-depth features extracted by CNN and BP.

## 4. Discussion and Conclusions

In this study, a novel hybrid system that combines an optimized CNN model with an SVR model was proposed, where CNN was used as a trainable feature extractor and SVR was performed as the regression operator. We demonstrated that a hybrid CNN-SVR model could effectively exploit feature interactions from feed-forward directions to learn more in-depth features of ECG and PPG signals and their corresponding SBP and DBP. In terms of *MAE* ± *STD*, the model achieved an overall accuracy of 1.23 ± 2.45 mmHg for SBP and 3.08 ± 5.67 for DBP, all of which comply with the accuracy criteria of the AAMI SP10 standard.

The proposed method can be efficiently utilized in an on-device application owing to the flexibility of CNN architecture to deal with various inputs such as ECG and PPG signals. Due to its calibration-free and unsupervised feature learning ability from the collected signal, the proposed method has high prospects for application in wearable BP monitoring devices. Moreover, there is no need for special conditions or additional algorithms for feature extraction, making it very cost-effective. Managing and predicting hypertension with a combination of the deep learning method and wearable technology could be a significant step forward in the treatment of cardiovascular disease. Thus, this model could play a significant role in continuous BP monitoring and early diagnosis of hypertension, consequently decreasing the rate of morbidity and mortality associated with CVD.

We assume that increasing the training sample size might affect the accuracy of the BP model measurement. Future work should investigate whether a new BP estimation model structure needs to be developed for a high-sampling-rate database. From a wearable application perspective, the average BP value and age of the subject in the MIMIC III database were higher than the total average of the general public, as these data were obtained from patients in ICUs. To improve the performance of this model, further experiments could be conducted using data collected through consultations in clinical environments with medical experts. 

## Figures and Tables

**Figure 1 sensors-23-01259-f001:**
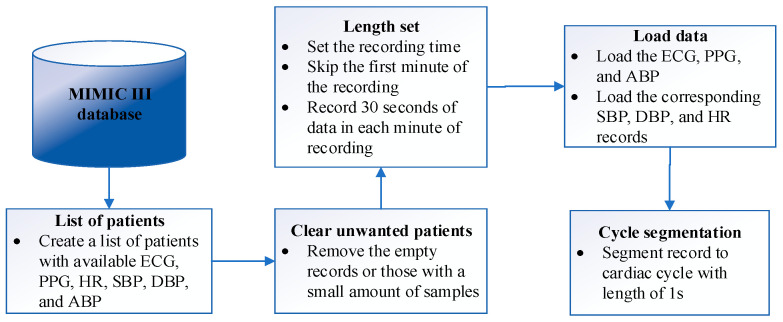
The steps to collect data for each patient.

**Figure 2 sensors-23-01259-f002:**
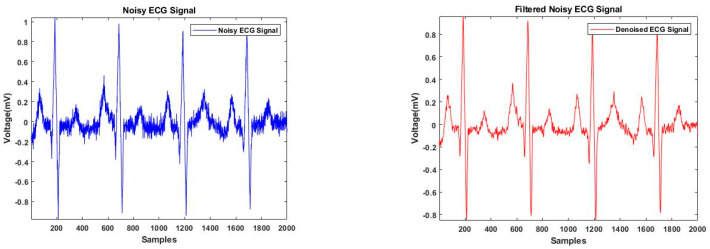
Noisy and denoised ECG signal.

**Figure 3 sensors-23-01259-f003:**
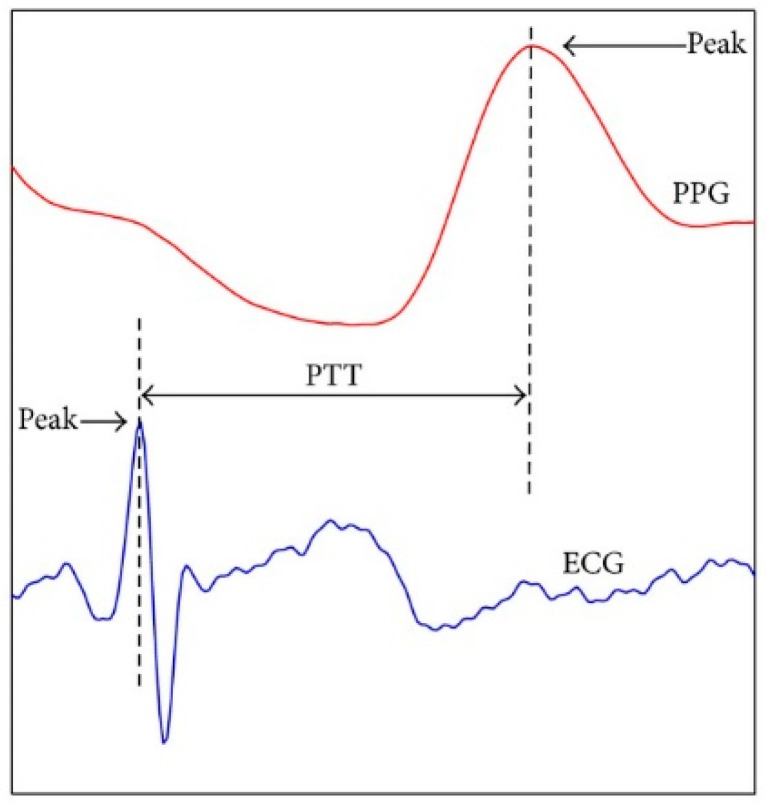
Graphical view of the *PTT* calculation.

**Figure 4 sensors-23-01259-f004:**
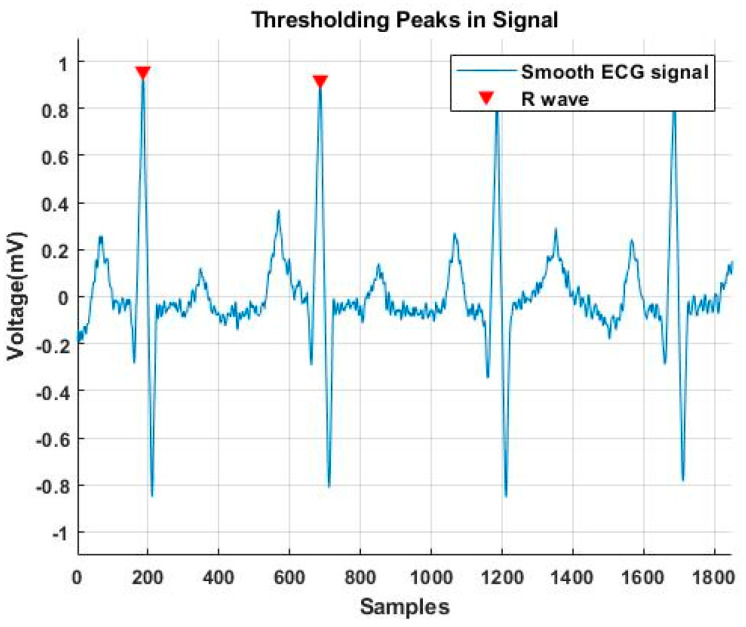
A sample of R-wave detection by applying the pick detection algorithm to an ECG signal.

**Figure 5 sensors-23-01259-f005:**
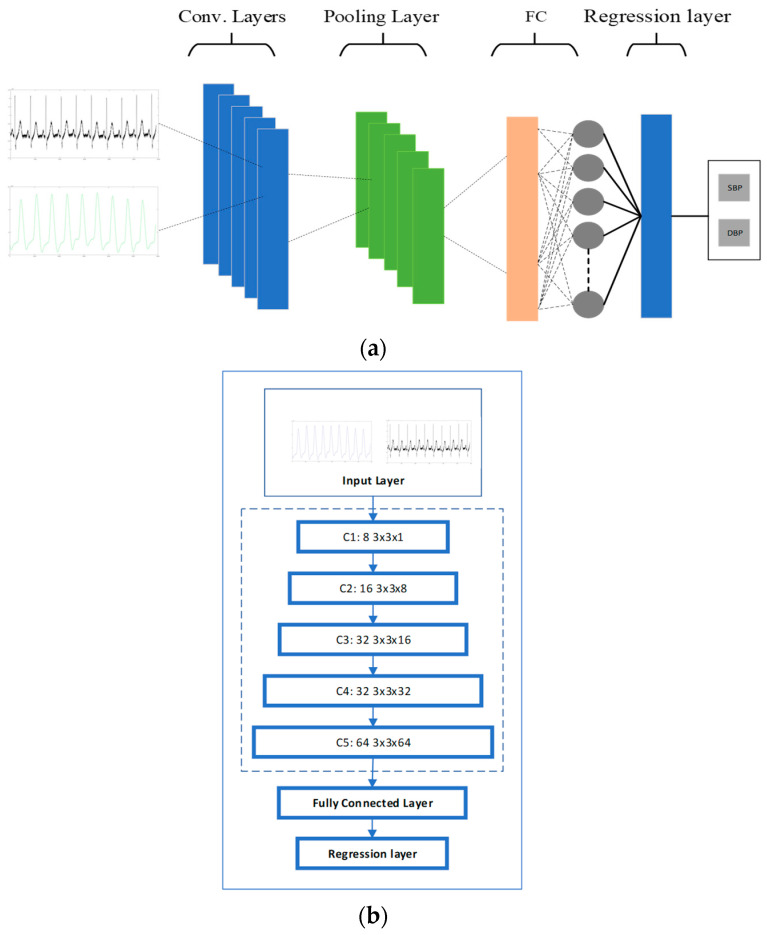
(**a**) CNN structure; (**b**) CNN layers.

**Figure 6 sensors-23-01259-f006:**
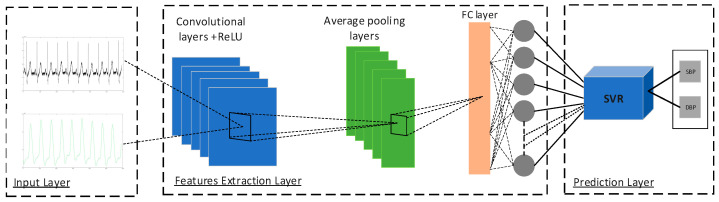
Structure of the CNN-SVR model.

**Figure 7 sensors-23-01259-f007:**
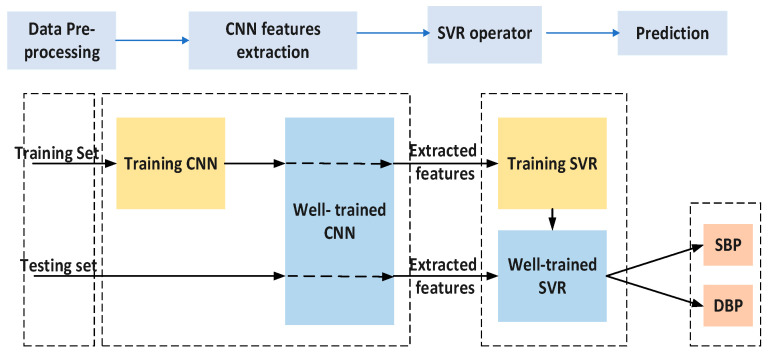
The implementation process of the Hybrid CNN-SVR model.

**Figure 8 sensors-23-01259-f008:**
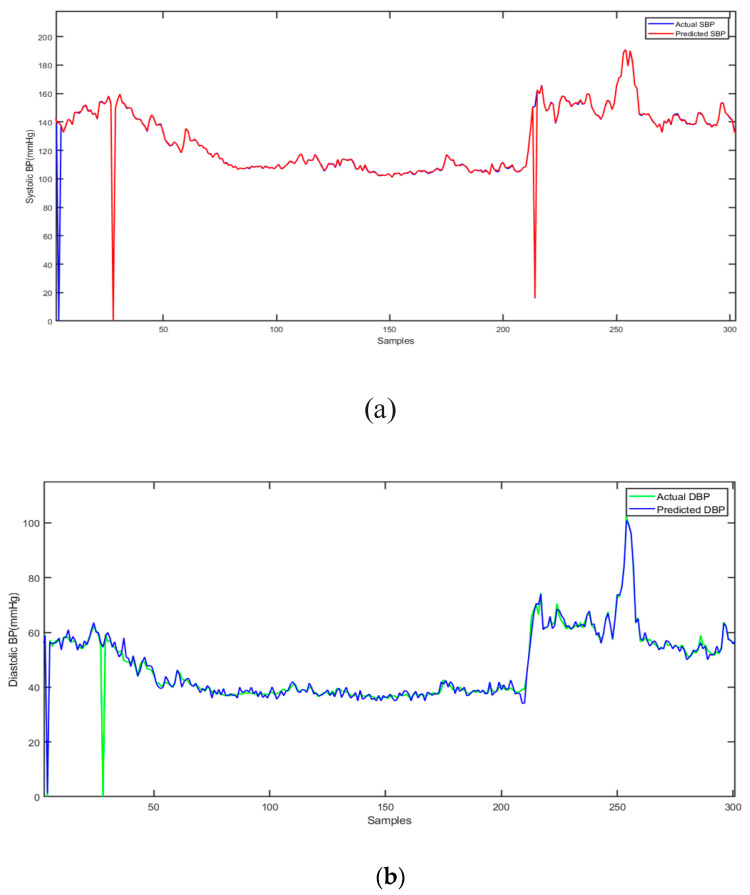
Comparison between (**a**) actual (blue) and predicted (red) SBP; (**b**) actual (green) and predicted (blue) DBP.

**Table 1 sensors-23-01259-t001:** Performance evaluation based on the AAMI standard.

		*MAE*	Standard Deviation	Subjects
AAMI standard	SB, DBP	≤5 (mmHg)	≤8 (mmHg)	≥85
Hybrid **CNN-SVR Model**	SBP	1.23	2.45	120
	DBP	3.08	5.67	

**Table 2 sensors-23-01259-t002:** Comparison of hybrid CNN-SVR BP model performance with other works in terms of methodology, database, use of engineered features, and estimation error.

Model	Number of Subjects	Engineered Feature	*MAE* SBP (mmHg)	*MAE* DBP (mmHg)
**Classical ML [38]**	MIMIC-II, 1000 subjects	Yes	11.17	5.35
**ResNet [39]**	MIMIC-III, 510 subjects	Yes	9.43	6.88
**GA-SVR [40]**	MIMIC-III, 772 waveforms	Yes	3.27	1.16
**Regression tree [41]**	Queensland, 32 subjects	Yes	4.82	3.25
**ELM [42]**	MIMIC-II, 4254 records	Yes	4.25	3.95
**ANN [43]**	MIMIC-II, 90 subjects	Yes	4.02	2.27
**CNN [44]**	Unspecified, 62 Subjects	No	9.61	6.73
**CNN [45]**	MIMIC-II, 379 Subjects	No	9.30	5.12
**Hybrid CNN-SVR**	MIMIC-III, 120 subjects	No	1.23	3.08

## Data Availability

The data in this study were obtained from the Medical Information Mart for Intensive Care (MIMIC-III) Waveform.

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
