# Peer review of "Hybrid CNN-SVR Blood Pressure Estimation Model Using ECG and PPG Signals"

_sensors, 2023, doi:10.3390/s23031259_

Round 1

Reviewer 1 Report

The authors present a novel CNN-SVR based method to predict the blood pressure by using the ECG and PPG signals. The manuscript is well written, it is easy to follow, and it presents minor typos. The main comments about this article are the following ones:

1.       More information about the ECG signals, i.e., number of leads used, is needed.

2.       I encourage you the use of well-known R-wave detectors such as the one presented in Physionet webpage. This change may improve the overall behavior of your method.

3.       More information about the input space is needed, how many leads are used?, the signals (ECG and PPG) are concatenate vertically or horizontally?, etc.

4.       In line 314, the reference is missed.

Author Response

Response to Reviewer 1 Comments

Point 1: More information about the ECG signals, i.e., number of leads used, is needed. 

Response 1: Five ECG signals from leads I, II, III, AVR, and V are available during portions of some ECG records in the MIMIC III waveform database. However, these five ECG signals are not all available simultaneously. We employed the ECG lead AVR in this study due to better signal quality and availability in most records. We also add this statement to the manuscript (lines 102 to 107).

Point 2:       I encourage you the use of well-known R-wave detectors such as the one presented in Physionet webpage. This change may improve the overall behavior of your method. 

Response 2: Thanks for your suggestion. We think the proposed R-wave detection accurately detects the R-wave in the selected signals. However, due to the time limitation, we cannot revise all experiments with a new R-wave detector and will consider your advice in our future work.

Point 3:        More information about the input space is needed, how many leads are used?, the signals (ECG and PPG) are concatenate vertically or horizontally?, etc. 

Response 3: We used lead AVR for ECG and PLETH for PPG signal in the MIMIC III waveform database as parallel input to the network and concatenated them vertically (lines 347 to 349).

Point 4:        In line 314, the reference is missed

Response 4: The reference has been added for Equation ‎2 (line 309).

Reviewer 2 Report

Writing:

- Typo in Page 3, Lines: 94-95, 

- It seems like there are some statements that should be in the introduction, but placed in methodology. 

- The information about PTT, CNN,  should be mentioned earlier in Introduction rather than in Methodology.

- Page 9,  "The non-linear model is shown in Equation Error! Reference source 314 not found.:"

- Page 5, Line 181, formatting problem.

- "The ECG signal (considered to be the most important component in BP estimation... " please have references for this statement.

Technical:

- On Page 2, Lines 86-87, "The ECG and PPG waveforms specified as AVR and PLETH were loaded to the MATLAB environment as a matrix with two labelled columns." This AVR is the ECG lead used in the calculation? Most likely, if the AVR lead signal, the R facing downward. This has conflicting statement with statement of "The R-peak of ECG signals was detected by thresholding peaks above 0.5 mV" given in Page 5 Lines 165-166. Please check.

- On Page 3 Line 92, ".. As each cardiac cycle interval was considered to be 0.6s to 1s,...". Please have references for this.

- On Page 4, Line 161, Figure 2, you have the noisy and clean ECG signals. Please plot the FFT for these corresponding signal to show the effectivity of the filtering technique in frequency domain.

- On Page 4, Lines 142-143, "The 142 source of these artifacts is often respiration and usually lies between 0.15 and 0.2 Hz" please have references to support the statement. Also, did you filter the PPG? how about the cutoff frequencies for this PPG signal?

- On Page 5, Line 167, "..the minimum peak distance was set by 125 samples which were equal to one cardiac cycle in this study". How to deal with the patients having the >60 BPM. Some R peaks will be discarded? Moreover, cardiac cycle, P-QRS-T, in some cases happened in less than 1 second. Please check the dataset.

- I am a bit confused, may I know what is the input of the CNN? the continuous ECG and PPG given in Fig. 4? or the PTT? because you gave the R peak detection in Fig. 3 and also a lot of information related to PTT as well? If you use directly ECG and PPG signals, I think it is better to reduce the information for PTT.

- Are you using 1D or 2D CNN? Which deep learning framework is used here? Keras(TensorFlow), Pytorch, or others? if in Keras, you can use "plot_model" function. It would be easier to explain your model's structure

- If you use the PTT for the input, please plot the ECG and the PPG at one figure, then we can see the interval between the R peaks and the PPG peaks for PTT. 

- It would be great if you put the comparative study, Table 2, in discussion part.

Author Response

Response to Reviewer 2 comments

Point 1: Typo in Page 3, Lines: 94-95,  

Response 1: The line is corrected to "A summary of the above steps is illustrated in Figure 1" (line 114).

Point 2:  It seems like there are some statements that should be in the introduction, but placed in methodology.

Response 2: We have incorporated this comment by moving some statements from the Methodology to the Introduction (lines 42 to 56).

  

Point 3:  The information about PTT, CNN,  should be mentioned earlier in Introduction rather than in Methodology.

Response 3: The pointed information has been removed or moved to the introduction section (lines 45 to 51).

Point 4:  Page 9,  "The non-linear model is shown in Equation Error! Reference source 314 not found.:" 

Response 4: Addressed (line 39).

Point 5:  Page 5, Line 181, formatting problem

Response 5: The format of this line is Corrected (line 179).

Point 6:  "The ECG signal (considered to be the most important component in BP estimation... " please have references for this statement. 

Response 6:  In the first paragraph of 2.2, the statement is modified to:

"The ECG signal is one of the crucial inputs to the BP estimation algorithm, and a clean and pre-processed ECG signal will increase the accuracy of BP measurement performance. (lines 122 to 124)"

Point 7:  On Page 2, Lines 86-87, "The ECG and PPG waveforms specified as AVR and PLETH were loaded to the MATLAB environment as a matrix with two labelled columns." This AVR is the ECG lead used in the calculation? Most likely, if the AVR lead signal, the R facing downward. This has conflicting statement with statement of "The R-peak of ECG signals was detected by thresholding peaks above 0.5 mV" given in Page 5 Lines 165-166. Please check. 

Response 7: We employed AVR lead for ECG and PLETH for PPG and used MATLAB to extract the BP information from these two signals. In the case of the R signal facing down, the inverted signal can be applied to the R-peak detection to avoid any false R-peak detection. More information about ECG signals is provided in the database section (lines 76 to 120).

Point 8: On Page 3 Line 92, ".. As each cardiac cycle interval was considered to be 0.6s to 1s,...". Please have references for this. 

Response 8: We addressed this comment on page 3 by rewording the statement to: "As each cardiac cycle interval was considered to be 0.6s to 1s, the signals were segmented into the cardiac cycle with a length of 1s as one beat [13] (line 113).

Point 9:  On Page 4, Line 161, Figure 2, you have the noisy and clean ECG signals. Please plot the FFT for these corresponding signal to show the effectivity of the filtering technique in frequency domain.

Response 9: We appreciate the comment. The filtering technique used in this study has a low-pass effect; therefore, we did not confirm its efficiency in the frequency domain. Shape preservation in the time domain (typical of polynomial filters) is more important in this application, as seen in Figure 2 (line 144).

Point 10: On Page 4, Lines 142-143, "The 142 source of these artifacts is often respiration and usually lies between 0.15 and 0.2 Hz" please have references to support the statement. Also, did you filter the PPG? how about the cutoff frequencies for this PPG signal? 

Response 10: Reference [14] supports this statement (line 127). With regards to the other question, it should be clarified that we didn't pre-process the PPG signal and used a normalization process for this signal. A window was selected with the R-peak positioned at the centre point of the window for resampling the cardiac intervals and the PPG signal.

Point 11:  On Page 5, Line 167, "..the minimum peak distance was set by 125 samples which were equal to one cardiac cycle in this study". How to deal with the patients having the >60 BPM.

Some R peaks will be discarded? Moreover, cardiac cycle, P-QRS-T, in some cases happened in less than 1 second. Please check the dataset. 

Response 11: According to the MIMIC III database, the sampling frequency is 125 Hz, corresponding to one cardiac cycle of 60 BPM. We considered the ECG signals with a heart rate of greater or less than 60 BPM by locating the R-wave in the centre of the window and eliminating samples from the tail of the signal for cases of less than 60 BPM or zero padding for cases greater than 60 BPM.  

Point 12:  I am a bit confused, may I know what is the input of the CNN? the continuous ECG and PPG given in Fig. 4? or the PTT? because you gave the R peak detection in Fig. 3 and also a lot of information related to PTT as well? If you use directly ECG and PPG signals, I think it is better to reduce the information for PTT. 

Response 12: We used the ECG and PPG signals directly collected from lead AVR and PELTH. The network is expected to consider the key signal points related to the PTT in the training phase and improve the accuracy. Some parts of the manuscript on information related to PPT have been deleted in the revised version to address this comment.

Point 13:  Are you using 1D or 2D CNN? Which deep learning framework is used here? Keras(TensorFlow), Pytorch, or others? if in Keras, you can use "plot_model" function. It would be easier to explain your model's structure 

Response 13: We used MATLAB environment and its deep learning toolbox. We are using 2D CNN with two inputs (one dimension is time, and the other is for signal source, i.e. ECG/PPG).

Point 14:  If you use the PTT for the input, please plot the ECG and the PPG at one figure, then we can see the interval between the R peaks and the PPG peaks for PTT.  

Response 14: A graphical view of the PTT calculation has been provided in Figure 3 (line 151).

Point 15:  It would be great if you put the comparative study, Table 2, in discussion part. 

Response 15: All the previous work included in table 2 are for comparison purpose due to using the same database, AI technologies, and hybrid models (line 429).

Round 2

Reviewer 2 Report

I have no more question.